# Test–Retest Reliability, Parallel Test Reliability, and Internal Consistency of Balance Assessments in Young Healthy Adults

**DOI:** 10.3390/jfmk10040455

**Published:** 2025-11-20

**Authors:** Teni Steingräber, Leon von Grönheim, Jana Wienecke, Rieke Regel, Christoph Schütz, Thomas Schack, Jitka Veldema

**Affiliations:** 1Faculty of Psychology and Sports Science, Bielefeld University, 33615 Bielefeld, Germany; teni.steingraeber@uni-bielefeld.de (T.S.); leonvongroenheim@gmx.de (L.v.G.); rieke.regel@uni-bielefeld.de (R.R.); christoph.schuetz@uni-bielefeld.de (C.S.); thomas.schack@uni-bielefeld.de (T.S.); 2Department of Exercise and Health, Paderborn University, 33098 Paderborn, Germany; jana.wienecke52@googlemail.com

**Keywords:** balance assessments, Y-Balance Test, Single-Leg Landing Test, Single-Leg Squat Test, force plate, centre of pressure, test–retest reliability, parallel test reliability, internal consistency, young healthy adults

## Abstract

**Objectives:** An objective evaluation of balance abilities is crucial in research, rehabilitation, sport, and daily life. With ongoing technical advancements, the number of innovative evaluation tools is continuously increasing. This study assessed the methodological quality of various differential balance assessments in young, healthy adults. **Methods:** Two technically sophisticated balance assessments using a force plate (Single-Leg Landing Test, Single-Leg Squat Test) and a conventional balance assessment using a simple test kit (Y-Balance Test) were applied to 42 students at two different time points. Test–retest reliability, parallel test reliability, and internal consistency were evaluated for each test and item. **Results:** All tests and (almost) all items showed excellent to acceptable test–retest reliability. In all tests, internal consistency was detected for only some items, while the other items were internally inconsistent. Only a small proportion of tests and/or their items demonstrated acceptable parallel test reliability. The balance performance of the right and left legs showed excellent or good reliability for each item. **Conclusions:** Significant test–retest reliability and consistency between right and left leg performance suggest good methodological quality of the assessments. The lack of parallel test consistency aligns with previous studies that emphasise the multi-faceted nature of balance tasks, suggesting that balance ability is task-specific rather than a “general ability”. Future studies should investigate and compare the biological and neural backgrounds of differential balance tasks to provide further insights into this topic.

## 1. Introduction

Balance is a fundamental motor skill that plays a critical role in daily life activities, injury prevention, rehabilitation, and sports performance [1,2,3]. Therefore, its objective evaluation is of significant interest. Numerous assessments are used to measure balance control in research and in clinical and sports practice [4,5,6,7]. Several studies have highlighted the limited parallel test consistency by using two or more different balance assessments applied to the same cohort [8,9,10,11,12,13,14]. For example, only weak consistencies have been found between (i) balance recovery after one jump landing, (ii) balance remaining during a single-leg stance, (iii) perturbation compensation during a single-leg stance, and (iv) balance recovery following a simulated forward fall in young adults [8]. Similarly, no relationships were detected between (i) balance remaining during bipedal step stance, (ii) perturbation compensation during bipedal step stance, and (iii) balance remaining during single-leg stance during reach movements of the contralateral leg in children [9]. These findings raise questions regarding the validity of the balance assessment. In contrast, the large variability indicates the multi-faceted nature of balance tasks and the wide array of mechanisms involved. Initially, a simple differentiation between static and dynamic balance was made. However, a recent systematic review and meta-analysis differentiated between (i) dynamic steady state, (ii) static steady state, and (iii) proactive balance [10] and reported only small-sized correlations between all balance types across the lifespan [10]. Similarly, the Balance Evaluation System Test—developed to reliably detect balance deficits in several patient cohorts—further differentiates six balance control systems: (i) biomechanical constraints, (ii) stability limits/verticality, (iii) anticipatory postural adjustments, (iv) postural responses, (v) sensory orientation, and (vi) stability in gait [12]. Deficits in one category are not necessarily associated with deficits in the remaining categories [12].

The study aims to provide evidence in this field and prove and compare the external as well as internal reliability of three balance assessments. A commonly used test, (i) the Y-Balance Test [13,14] is compared with two technically sophisticated centre of pressure (COP) evaluations through a force plate during the (ii) Single-Leg Landing Test [14,15] and (iii) Single-Leg Squat Test [14,15]. Such assessments have the potential to provide nuanced insight into balance control (which traditional tests are incapable of) and become more common in upcoming years. The Y-Balance Test is based on the measurement of the maximum reach distance of the lower extremity in three directions, (i) anterior, (ii) posteromedial, and (iii) posterolateral, while standing on the opposite leg [13,14]. This simple and low-cost test was developed as a more standardised and reliable version of the Star Excursion Balance Test [13,16]. COP recording movements using a force plate can be performed during various balance-related tasks. COP movement trajectory length/area, COP movement velocity/frequency, and time required for stabilisation were determined during the wide stance, tandem stance, one-leg stance, single-leg squat, and single-leg landing with open/closed eyes in previous studies [8,14,15,17]. Our study applies a Single-Leg Squat Test and a Single-Leg Landing Test with open eyes.

## 2. Methods

### 2.1. Study Design

In this observational study, balance ability was evaluated using three different assessments: the Y-Balance Test, Single-Leg Landing Test, and Single-Leg Squat Test (Figure 1a–c). Assessments were conducted at two different time points, with at least 48 h between them, by the same examiner. The participants had the opportunity to practice the movement prior to testing. After them, both the right and left legs were tested in a randomised order for each participant. Following the tests, test–retest reliability, internal consistency, and parallel test reliability were assessed. This study was conducted in accordance with the Declaration of Helsinki and was approved by the Ethics Committee of Bielefeld University (approval no. 2022-043).

### 2.2. Participants

A total of 42 healthy students were included in this study (age 25.1 ± 3.2 years; height 1.76 ± 0.08 m; weight 73.5 ± 11.4 kg; 19 females, 23 males; 36 right-footed, 6 left-footed). The student’s preferred foot to kick a ball was considered dominant [18].

### 2.3. Balance Tests

#### 2.3.1. Y-Balance Test

The maximal reach of the free lower leg in the (a) anterior, (b) posterolateral, and (c) posteromedial directions was measured during a single-leg stance on the opposite leg using a standardised test kit (FMS, Chatham, VA, USA) (Figure 1a) [13,14]. The arm position was not specified. Five attempts were conducted for each leg and direction, and the mean value of the five trials was used for the analysis. A greater reach distance indicates a better balance ability.

#### 2.3.2. Single-Leg Landing Test

The participants were instructed to perform a forward jump (covering 50% of their body height) and land on a single limb positioned on a force plate, with their hands resting on their hips (Figure 1b) [14,15]. The COP movement in the horizontal plane was recorded, and the following variables were determined: (i) the COP movement trajectory length [19] (Figure 1d), (ii) COP movement trajectory area [19] (Figure 1e), and (iii) time needed for stabilisation [20] (Figure 1f). Five trials were performed for each leg, and the mean values were used for the analysis. A smaller COP trajectory length and area and shorter time needed for stabilisation indicate better balance. A force plate (AMTI, Watertown, MA, USA) was used to collect the ground reaction force data at a sampling rate of 1000 Hz. The analogue signals from the force plate were amplified using an AMTI amplifier (Watertown, MA, USA) and recorded via a Vicon MX analogue to digital interface unit (Vicon Motion Systems Ltd., Yarnton, Oxford, UK). The data was filtered using a fourth-order zero-phase Butterworth digital filter [19] with an estimated optimum cut-off frequency of 12.53 Hz [20].

#### 2.3.3. Single-Leg Squat Test

Participants performed five consecutive squats to a depth equivalent to 10% of their body height during a single-leg stance on a force plate (as previously described), with their hands resting on their hips [14,15] (Figure 1c). The COP movement in the horizontal plane was recorded, and the following variables were determined: (i) COP movement trajectory length (Figure 1d) and (ii) COP movement trajectory area (Figure 1e). One attempt was conducted for each leg. A smaller COP trajectory length and area indicate better balance ability.

### 2.4. Statistical Analysis

The data collected in this study were analysed using the SPSS software package, version 27 (International Business Machines Corporation Systems, IBM, Ehningen, BW, Germany). Intraclass correlation coefficients (ICCs) were calculated to assess the test–retest reliability for all tests and their items. Higher ICCs indicate better reliability (ICC ≥ 0.9 = excellent; 0.9 > ICC ≥ 0.75 = good; 0.75 > ICC ≥ 0.5 = moderate; ICC < 0.5 = poor) [21]. Cronbach’s α coefficients were calculated to assess (i) the test–retest reliability, (ii) internal consistency, and (iii) parallel test reliability [22,23] for all tests and their items. Higher Cronbach’s α values indicate better reliability and consistency (α ≥ 0.9 = excellent; 0.9 > α ≥ 0.8 = good; 0.8 > α ≥ 0.7 = acceptable; 0.7 > α ≥ 0.6 = questionable; 0.6 > α ≥ 0.5 = poor; α < 0.5 = unacceptable) [22,23].

## 3. Results

Table 1 presents an overview of the data (means and standard deviations) collected during the experiment.

### 3.1. Test–Retest Reliability

The test–retest reliability for all tests and their items is shown in Table 1. The Y-Balance Test and Single-Leg Landing Test demonstrated excellent or good test–retest reliability for both the total score and their items (with the exception of anterior direction items of the Y-Balance Test that showed moderate reliability only). In contrast, the Single-Leg Squat Test showed good test–retest reliability for COP movement trajectory length but poor reliability for the COP movement trajectory area.

### 3.2. Internal Consistency

The internal consistency of all tests and their items are presented in Table 2, Table 3 and Table 4.

#### 3.2.1. Y-Balance Test

Only the posterolateral and posteromedial direction items—but not the anterior direction item—showed consistency (good, acceptable, or questionable) with the total Y-Balance Test score (Table 2). Additionally, only the posteromedial and posterolateral direction items demonstrated mutual consistency (excellent, good, or acceptable). No acceptable consistency was observed for the anterior direction item concerning both remaining (posterolateral and posteromedial) direction items. The balance control of the right and left legs within each direction item (anterior, posterolateral, posteromedial) was consistent (excellent or good).

#### 3.2.2. Single-Leg Landing Test

The COP movement trajectory area and trajectory length items showed questionable to unacceptable consistency (Table 3). The time needed for stabilisation did not show acceptable consistency, either with the COP movement trajectory area item or with the COP trajectory length item. The balance control of both the right and left legs showed good consistency within each item (trajectory area, trajectory length, and time needed for stabilisation).

#### 3.2.3. Single-Leg Squat Test

The COP movement trajectory area and trajectory length items were not consistent (Table 4). However, the performance of the right and left leg within each item (trajectory area, length) was consistent (good, acceptable or questionable).

### 3.3. Parallel Tests Reliability

Of the variables assessed, only COP movement trajectory length demonstrated any degree of consistency across different test modalities (Table 5). The Y-Balance Test, COP trajectory area items of both the Single-Leg Landing Test and Single-Leg Squat Test, and the time needed for stabilisation item of the Single-Leg Squat Test showed no acceptable mutual consistency.

## 4. Discussion

This study assessed the methodological quality of three balance assessments—the Y-Balance Test, Single-Leg Landing Test and Single-Leg Squat Test—in young, healthy adults. The findings indicate that (i) almost all tests and their items demonstrated excellent to good test–retest reliability (except anterior direction items of the Y-Balance Test that showed moderate reliability and the COP movement trajectory area item of the Single-Leg Squat Test that showed poor reliability); (ii) all tests exhibited at least acceptable internal consistency only for a part of their items; (iii) only a small part of the tests and/or their items showed (at least) acceptable parallel test reliability; and (iv) the balance performance of the right and left leg within one item was generally consistent, with excellent or good reliability. Altogether, the technically advanced measurements of COP using the force plate (Single-Leg Landing Test, Single-Leg Squat Test) showed no superior methodological quality than a simple determination of reach distance (Y-Balance Test) using a test kit.

### 4.1. Y-Balance Test

A previous review reported excellent intra-rater reliability for the Y-Balance Test across the anterior, posteromedial, and posterolateral directions in healthy populations [24]. In contrast, our findings demonstrated excellent intra-rater reliability for the posteromedial and posterolateral but only acceptable reliability for the anterior direction. Similarly, internal consistency tests for posteromedial and posterolateral directions showed both good mutual consistency as well as good consistency with the overall Y-Balance Test score, while no relevant consistency was observed for the anterior direction. Several studies indicate that the anterior reach movement during the Y-Balance Test may be strongly determined by joint mobility and weakly by balance control through neural networks [25,26,27,28,29]. Ankle dorsiflexion [26,27,29], knee flexion [27,28] or hip extension [27,29] have been identified as relevant predictors of reached distance in the anterior direction, while posteromedial and posterolateral reach distance appear to be less affected by a joint’s movement range [25,26,27,28,29]. Similarly, intervention-induced improvement on the Y-Balance Test was found only for the posteromedial and posterolateral directions but not for the anterior direction in a neuromodulation study [14].

Our data did not show relevant consistency between the Y-Balance Test and the Single-Leg Landing Test and/or Single-Leg Squat Test items. Similarly, previous studies demonstrated the consistency of the Y-Balance Test with the Star Excursion Test (both have a similar design) [30] but have shown inconsistency with balance assessments that are based on widely differential approaches [9,14]. For example, the Functional Reach Test performance has correlated neither with stance on an unstable platform nor with bipedal stance in a children’s cohort [9]. Intervention-induced improvements were detected on the Y-Balance Test but not in the Single-Leg Landing Test and Single-Leg Squat Tests in a neuromodulation study with young adults [14].

### 4.2. Single-Leg Landing Test

Previous data on test–retest reliability of the Single-Leg Landing Test is highly variable [31,32]. Unacceptable consistency for time to stabilisation was detected in professional rugby union players [31], whereas good (within a single day) and excellent (between days) test–retest reliability was observed in a cohort of young healthy adults [32]. Our study showed an excellent test–retest reliability for both time to stabilisation and movement trajectory length items and a good reliability in the movement trajectory area size item. However, the internal consistency of the Single-Leg Landing Test was limited. Our data demonstrated only questionable consistency between COP movement trajectory length and trajectory area size items, and no consistency for the time to stabilisation items. Furthermore, comparisons with the Y-Balance Test and Single-Leg Squat Test showed only little similarities. Only the COP movement trajectory length of the Single-Leg Landing Test and Single-Leg Squat Test demonstrated questionable consistency. This suggests that the COP movement trajectory length may be the most valid item of the Single-Leg Landing Test. Our findings are consistent with a previous study reporting only weak correlations of the Single-Leg Landing Test with other balance assessments (Stance on Unstable Platform Test and Forward Falls Test) in young healthy adults [8].

### 4.3. Single-Leg Squat Test

To the best of our knowledge, the methodological quality of COP movement trajectories during the Single-Leg Squat Test has not previously been investigated. Our study addresses this gap in this field. Our data demonstrated good test–retest consistency for COP movement trajectory length, unacceptable consistency for COP movement trajectory area size, and no consistency between these two items. The parallel tests comparison showed questionable consistency between the Single-Leg Squat and Single-Leg Landing Tests only for COP movement trajectory length items.

## 5. Conclusions

Our findings on the reliability of balance assessments investigated are ambiguous. Satisfactory test–retest reliability as well as excellent consistency between the right and left leg points to good methodological quality across all assessments. Missing parallel test consistency between the Y-Balance Test, Single-Leg Landing Test and Single-Leg Squat Test aligns with previous studies suggesting that balance performance is task-specific rather than a “general ability” [8,9,10,11,12,13,14]. The inconsistencies observed between the single items within both the Single-Leg Landing Test and Single-Leg Squat Test in our study are surprising, especially given their conceptual similarities. The COP trajectory length seems to be more reliable than the remaining items in both tests.

The leading determinants of missing parallel tests consistencies may be interindividual differences in somatotypes, anatomical conditions or neural networks that impact the performance in differential balance tasks to different degrees and ways [33,34,35]. For example, a study revealed that a greater body weight and body height are predictors of better dynamic, but not static, balance in young athletes [33]. Another study demonstrated that abnormalities of the longitudinal arch of the foot (pronation or supination) are associated (in comparison to a normal foot) with worsened balance performance during a Flamingo Test (stand on one leg while the other leg is bent and lifted), but not during a Y-Balance Test [34]. A previous review of neuroimaging data indicates that the cerebellum and the brainstem play crucial roles during dynamic balance tasks, while static balance is strongly associated with frontal, occipital, and parietal regions [35]. Thus, interindividual differences in neural network may also cause the performance inconsistencies during different balance exercises. Existing EEG studies demonstrated for example, that bipedal standing under difficult conditions (soft surface, closed eyes) is associated not only with the greatest COP instability but also with significantly different EEG patterns at the parietal and central brain regions compared to bipedal standing under simple conditions (hard surface, opened eyes) [36,37]. More studies on this field are needed for a better understanding of multi-faceted balance control. Multiple cortical and subcortical regions, the brain stem, spinal cord networks, vestibular circuits, muscle spindles, and other structures are crucially involved in balance regulation [12,14,38,39,40,41]. The relationships between illness-induced disruptions within some of these regions and balance impairment during specific balance-related tasks have already been demonstrated in various patient populations [12].

## 6. Strengths and Limitations

This study examined the methodological quality of two previously underexplored balance evaluation tools—the COP-based Single-Leg Landing Test and Single-Leg Squat Test—alongside the well-established Y-Balance Test. Our findings help to bridge an existing gap in the literature and contributes to the development and application of reliable balance assessments. Whether a better familiarisation of both probands and examinators, as well as an inclusion of additional trials or participants, would lead to more reliable results is not certain. The waiving of normalization of the metrics by high may make a Teni Steingräbercomparison of our results with other studies on this field difficult.

## Figures and Tables

**Figure 1 jfmk-10-00455-f001:**
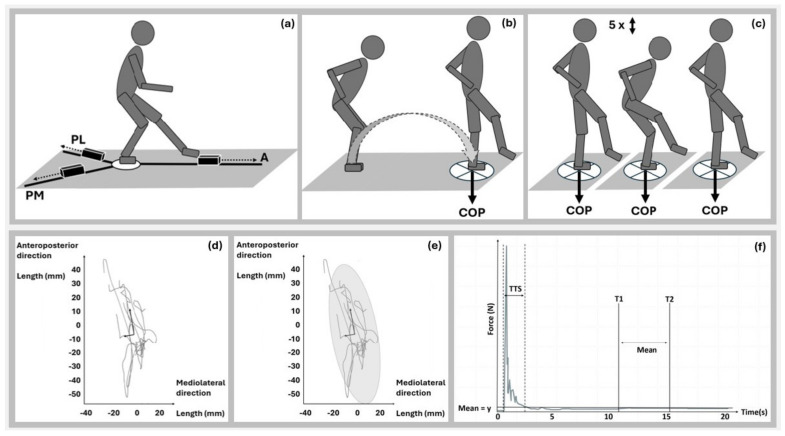
Y-Balance Test (**a**), Single-Leg Landing Test (**b**), Single-Leg Squat Test (**c**), COP movement trajectory length (COP path trajectory from initial ground contact until postural control regaining) (**d**), COP movement trajectory area (area of 95% confidence ellipse enclosing the COP movement trajectory) (**e**), time needed for stabilisation (time from initial ground contact until postural control regaining) (**f**). Time windows between T1 and T2 represent the reference value. Notes: A = anterior direction; COP = centre of pressure; N = Newton; PM = posteromedial direction; PL = posterolateral direction; s = second; T = time; TTS = time to stabilisation.

**Table 1 jfmk-10-00455-t001:** Test–retest (means, SD, reliability) of Y-Balance Test, Single-Leg Landing Test, and Single-Leg Squat Test.

	Test (Means and SD)	Retest (Means and SD)	Test–Retest Reliability (ICC)	95% Confidence Interval		Significance
				Lower Limit	Upper Limit	
**Y-Balance Test total (cm)**	521.88 ± 41.66	523.82 ± 5.15	**0.947**	0.900	0.972	<0.001
**Anterior direction (cm)**	113.42 ± 9.99	114.38 ± 15.14	0.745	0.518	0.865	<0.001
Left leg (cm)	56.73 ± 4.87	57.15 ± 6.29	**0.908**	0.825	0.952	<0.001
Right leg (cm)	56.48 ± 5.37	56.97 ± 7.10	**0.852**	0.721	0.922	<0.001
**Posterolateral direction (cm)**	208.93 ± 19.93	210.28 ± 23.85	**0.905**	0.824	0.949	<0.001
Left leg (cm)	104.02 ± 10.05	104.68 ± 11.92	**0.894**	0.803	0.943	<0.001
Right leg (cm)	104.91 ± 10.72	105.77 ± 12.64	**0.890**	0.796	0.941	<0.001
**Posteromedial direction (cm)**	202.44 ± 22.19	204.15 ± 25.58	**0.944**	0.895	0.970	<0.001
Left leg (cm)	102.00 ± 12.52	102.28 ± 12.89	**0.931**	0.871	0.963	<0.001
Right leg (cm)	101.23 ± 11.33	101.27 ± 12.96	**0.946**	0.898	0.971	<0.001
**Single-Leg Landing Test**						
**COP movement trajectory area size (mm^2^)**	3542 ± 831	3444 ± 747	**0.848**	0.707	0.921	<0.001
Left leg (mm^2^)	1720 ± 412	1724 ± 355	**0.833**	0.681	0.912	<0.001
Right leg (mm^2^)	1818 ± 464	1722 ± 458	**0.778**	0.572	0.884	<0.001
**COP movement trajectory length (mm)**	2340 ± 387	2263 ± 392	**0.904**	0.817	0.950	<0.001
Left leg (mm)	1151 ± 193	1127 ± 195	**0.900**	0.812	0.947	<0.001
Right leg (mm)	1180 ± 230	1140 ± 218	**0.852**	0.720	0.922	<0.001
**Time to stabilisation (ms)**	2.454 ± 0.318	2.435 ± 0.325	**0.913**	0.836	0.953	<0.001
Left leg (ms)	1.260 ± 0.166	1.236 ± 0.164	**0.839**	0.698	0.914	<0.001
Right leg (ms)	1.194 ± 0.171	1.199 ± 0.183	**0.912**	0.836	0.953	<0.001
**Single-Leg Squat Test**						
**COP movement trajectory area size (mm^2^)**	6609 ± 1618	6408 ± 2225	0.428	−0.082	0.697	0.043
Left leg (mm^2^)	3327 ± 1111	3209 ± 1230	0.462	−0.009	0.713	0.027
Right leg (mm^2^)	3245 ± 917	3198 ± 1266	0.334	−0.248	0.645	0.101
**COP movement trajectory length (mm)**	1979 ± 363	2029 ± 524	**0.865**	0.743	0.929	<0.001
Left leg (mm)	1015 ± 223	1054 ± 312	**0.785**	0.598	0.886	<0.001
Right leg (mm)	980 ± 187	998 ± 257	**0.816**	0.649	0.903	<0.001

**Notes: 0.9 ≤ ICC = excellent; 0.9 > ICC ≥ 0.75 = good;** 0.75 > ICC ≥ 0.5 = moderate; ICC < 0.5 = poor; COP = centre of pressure; cm = centimetre; mm = millimetre; ms = millisecond; SD = standard deviation.

**Table 2 jfmk-10-00455-t002:** Internal consistency (Cronbach’s α) of the Y-Balance Test.

	Total (cm)	Anterior Direction (cm)	Left Leg (cm)	Right Leg (cm)	Posterolateral Direction (cm)	Left Leg (cm)	Right Leg (cm)	Posteromedial Direction (cm)	Left Leg (cm)
**Anterior direction (cm)**	0.318								
Left leg (cm)	0.172	**0.972**							
Right leg (cm)	0.187	**0.896**	**0.954**						
**Posterolateral direction (cm)**	**0.826**	0.326	0.301	0.223					
Left leg (cm)	0.559	0.337	0.302	0.270	**0.872**				
Right leg (cm)	0.590	0.397	0.283	0.373	**0.893**	**0.939**			
**Posteromedial direction (cm)**	**0.854**	0.146	0.086	0.088	**0.889**	0.720	0.754		
Left leg (cm)	0.618	0.229	0.175	0.178	**0.842**	**0.859**	**0.886**	**0.888**	
Right leg (cm)	0.609	0.131	0.099	0.103	**0.818**	**0.859**	**0.875**	**0.882**	**0.961**

**Notes: 0.9 ≤ α = excellent; 0.9 > α ≥ 0.8 = good;** 0.8 > α ≥ 0.7 = acceptable; 0.7 > α ≥ 0.6 = questionable; 0.6 > α ≥ 0.5 = poor; α < 0.5 = unacceptable; cm = centimetre; mm = millimetre.

**Table 3 jfmk-10-00455-t003:** Internal consistency (Cronbach’s α) of the Single-Leg Landing Test.

	COP Movement Trajectory Area Size (mm^2^)	Left Leg (mm^2^)	Right Leg (mm^2^)	COP Movement Trajectory Length (mm)	Left Leg (mm)	Right Leg (mm)	Time to Stabilisation (ms)	Left Leg (ms)
**COP movement trajectory area size (mm^2^)**								
Left leg (mm^2^)	**0.853**							
Right leg (mm^2^)	**0.894**	**0.850**						
**COP movement trajectory length (mm)**	0.617	0.617	0.679					
Left leg (mm)	0.389	0.579	0.493	**0.853**				
Right leg (mm)	0.438	0.553	0.605	**0.897**	**0.870**			
**Time to stabilisation (ms)**	0.000	−0.001	0.000	0.000	0.000			
Left leg (ms)	0.000	0.000	0.000	0.000	0.000	0.000	**0.871**	
Right leg (ms)	0.000	0.000	0.000	0.000	0.000	0.000	**0.882**	**0.872**

**Notes: 0.9 ≤ α = excellent; 0.9 > α ≥ 0.8 = good;** 0.8 > α ≥ 0.7 = acceptable; 0.7 > α ≥ 0.6 = questionable; 0.6 > α ≥ 0.5 = poor; α < 0.5 = unacceptable; cm = centimetre; COP = centre of pressure; mm = millimetre; ms = millisecond.

**Table 4 jfmk-10-00455-t004:** Internal consistency (Cronbach’s α) of the Single-Leg Squat Test.

	COP Movement Trajectory Area Size (mm^2^)	Left Leg (mm^2^)	Right Leg (mm^2^)	COP Movement Trajectory Length (mm)	Left Leg (mm)
**COP movement trajectory area size (mm^2^)**					
Left leg (mm^2^)	**0.880**				
Right leg (mm^2^)	**0.830**	0.659			
**COP movement trajectory length (mm)**	−0.042	0.086	−0.117		
Left leg (mm)	0.022	0.106	−0.011	**0.884**	
Right leg (mm)	−0.059	−0.035	−0.059	**0.849**	0.777

**Notes: 0.9 ≤ α = excellent; 0.9 > α ≥ 0.8 = good;** 0.8 > α ≥ 0.7 = acceptable; 0.7 > α ≥ 0.6 = questionable; 0.6 > α ≥ 0.5 = poor; α < 0.5 = unacceptable; cm = centimetre; COP = centre of pressure; mm = millimetre.

**Table 5 jfmk-10-00455-t005:** Parallel test reliability (Cronbach’s α) of (i) Y-Balance Test, (ii) COP movement trajectory area size, (iii) COP movement trajectory length, (iv) time to stabilisation during Single-Leg Landing Test, (v) COP movement trajectory area size, and (vi) COP movement trajectory length during Single-Leg Squat Test.

	Y-Balance Test (cm)	Single-Leg Landing Test	COP Movement Trajectory Area Size (mm^2^)	GOP Movement Trajectory Length (mm)	Time to Stabilisation (ms)	Single Leg-Squat Test	COP Movement Trajectory Area Size (mm^2^)
**Single-Leg Landing Test**							
COP movement trajectory area size (mm^2^)	0.017						
GOP movement trajectory length (mm)	0.021		0.618				
Time to stabilisation (ms)	−0.008		0.000	−0.046			
**Single-Leg Squat Test**							
COP movement trajectory area size (mm^2^)	0.014		0.215	−0.116	0.000		
COP movement trajectory length (mm)	−0.007		0.291	0.646	0.000		−0.116

**Notes: 0.9 ≤ α = excellent; 0.9 > α ≥ 0.8 = good;** 0.8 > α ≥ 0.7 = acceptable; 0.7 > α ≥ 0.6 = questionable; 0.6 > α ≥ 0.5 = poor; α < 0.5 = unacceptable; cm = centimetre; COP = centre of pressure; mm = millimetre.

## Data Availability

The original contributions presented in this study are included in the article. Further inquiries can be directed to the corresponding author.

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
