# Peer review of "Test–Retest Reliability, Parallel Test Reliability, and Internal Consistency of Balance Assessments in Young Healthy Adults"

_jfmk, 2025, doi:10.3390/jfmk10040455_

Round 1
Reviewer 1 Report
Comments and Suggestions for Authors
Stable balance is a prerequisite for almost all forms of locomotion, therefore its assessment is crucial to understand its underlying mechanisms and how it changes in the case of aging, disease, injury, etc. Many balance assessment types exist, but often their reliability and consistency metrics go unpublished. Valid, consistent, and reliable tests must be selected in order to most accurately assess balance, without complicating interpretation due to unknown confounders. This paper reports on these very features in the context of 1 common balance assessment, and 2 less common, more complex balance assessments.
Specific comments:
- introduction: it is not clear why the more "sophisticated" balance tests were selected. Justification should be included in the discussion, i.e. these tests are becoming more common in recent years, or they provide nuanced insight into balance control that other tests are incapable of. Please expand.
- the last sentence of the last paragraph of the introduction needs clarification. Currently, it reads as if the study will include these measures, where, in reality, the sentence seems to actually call upon previous works and their features of interest.
- methods: for the single leg squat test - were the mean values of measures across the 5 squats used for analysis? Please clarify. Also, for such "sophisticated" tests, were the participants given an opportunity to practice the movement? This should be noted.
- methods: treatment of COP data is not mentioned. Sampling frequency, filtering procedure, etc. Please amend.
- results: 3.2.2. "questionable" should be replaced with a more accurate term according to the authors' statistical section. The results should be objective.
- discussion (first paragraph): 'excepting' --> except
- discussion: the authors mention that somatotypes or anatomical conditions may play a role in balance performance. Were there own measures normalized by height, as in common in COP-based metrics? If not, considering reanalyzing the data with this in mind. Otherwise, this should be mentioned as a limitation.
- discussion: several studies also now allow for simultaneous recording of brain activity and postural sway...which appear to give us critical insight into how nervous system controls balance. Refs: Legrand, T., Mongold, S. J., Muller, L., Naeije, G., Ghinst, M. V., & Bourguignon, M. (2024). Cortical tracking of postural sways during standing balance. Scientific Reports, 14(1), 1-13. Tse, Y. Y. F., Petrofsky, J. S., Berk, L., Daher, N., Lohman, E., Laymon, M. S., & Cavalcanti, P. (2013). Postural sway and rhythmic electroencephalography analysis of cortical activation during eight balance training tasks. Medical science monitor: international medical journal of experimental and clinical research, 19, 175.
- limitations: the results are not generalizable to older individuals, where balance tests are arguably more common. The same applies to those with injuries or diseases. Whether familiarization is necessary to achieve better reliability is not known, as is the inclusion of additional trials.
Author Response
Dear reviewer,
thank you very much for the time and effort to review our manuscript.
We performed a revision in line with your comments.
thank you for your efforts

Reviewer 2 Report
Comments and Suggestions for Authors
The research has achieved its goal and the results obtained can help specialists to better choose a balance test depending on the established objective. The paper confirms some data present in the specialized literature without however elucidating very clearly the factors that influence balance.

Author Response
Dear reviewer, thank you very much for the positive evaluation of our manuscript.
Reviewer 3 Report
Comments and Suggestions for Authors
Balance is a fundamental motor skill that plays a critical role in daily living, injury prevention, rehabilitation, and athletic performance.
The aim of the study was to compare the internal and external reliability of three balance assessments. A widely used test, (i) the Y balance test, was compared with two technically challenging force platform assessments of the center of pressure (COP) during (ii) a single-leg landing test and (iii) a single-leg squat test.
The article requires revision. A clearer description of the research methodology is needed.
In the Methodology section, you must specify the equipment used to measure balance and the parameters measured. It is necessary to indicate the principle of the comparison carried out, what measurement was used for the test and retest. What do the authors mean by the concept of measurement quality?
The results should be presented more clearly. Provide an introductory sentence for Table 1. Give a title to Table 1.
As the authors noted, the study sample size was small, so they did not obtain clear results for some measurements. A larger sample size would be necessary to make the results more conclusive. Determine the sample size required to obtain reliable results.
Author Response
Dear reviewer,
thank you very much for the time and effort to review our manuscript
We performed a revision in line with your comments.

Round 2
Reviewer 3 Report
Comments and Suggestions for Authors
The manuscript has been improved. However, some comments were not taken into account.
Provide an introductory sentence for Table 1. Give a title to Table 1. These changes are not reflected in the new version.
Author Response
Comment: Provide an introductory sentence for Table 1. Give a title to Table 1. These changes are not reflected in the new version.
Answer: Dear reviewer, the manuscript was revised in accordance with your comment:
Table 1: Test-retest (means, SD, reliability) of Y Balance Test, Single Leg Landing Test and Single Leg Squat Test.